# Risk Factors for Perioperative Urinary Tract Infection After Living Donor Kidney Transplantation Characterized by High Prevalence of Desensitization Therapy: A Single-Center Analysis

**DOI:** 10.3390/jcm14176102

**Published:** 2025-08-28

**Authors:** Shingo Nishimura, Shota Inoue, Takanori Sekito, Ichiro Tsuboi, Moto Tokunaga, Kasumi Yoshinaga, Yuki Maruyama, Yosuke Mitsui, Tomoaki Yamanoi, Tatsushi Kawada, Risa Kubota, Takuya Sadahira, Yusuke Tominaga, Takehiro Iwata, Satoshi Katayama, Kensuke Bekku, Kohei Edamura, Koichiro Wada, Yasuyuki Kobayashi, Motoo Araki

**Affiliations:** 1Department of Urology, Okayama University Graduate School of Medicine, Dentistry and Pharmaceutical Sciences, 2-5-1, Shikata-cho, Kita-ku, Okayama 700-8558, Japan; inouroshota@gmail.com (S.I.); ichiro.tsuboi0810@gmail.com (I.T.); pzxh8r1r@s.okayama-u.ac.jp (K.Y.); m28949825@gmail.com (Y.M.); ppv86vc7@s.okayama-u.ac.jp (T.Y.); p7ez8zvc@s.okayama-u.ac.jp (T.K.); pnrs4ypc@s.okayama-u.ac.jp (T.S.); p3uq1s4o@s.okayama-u.ac.jp (Y.T.); pfty8bwj@s.okayama-u.ac.jp (T.I.); pzgr6t3w@s.okayama-u.ac.jp (S.K.); gmd421030@s.okayama-u.ac.jp (K.B.); pa7k8q7o@s.okayama-u.ac.jp (K.E.); motoosh@md.okayama-u.ac.jp (M.A.); 2Department of Inflammation and Immunity, Lerner Research Institute, Cleveland Clinic, Cleveland, OH 44195, USA; t.sekito410@gmail.com (T.S.); eat.the.lobster@gmail.com (Y.M.); 3Department of Urology, NHO Okayama Medical Center, Okayama 701-1192, Japan; tokunaga10010@gmail.com (M.T.); m0r0s.0806.bye.gones@gmail.com (R.K.); 4Department of Urology, Shimane University Faculty of Medicine, Izumo 693-0021, Japan; wada-uro@med.shimane-u.ac.jp; 5Department of Urology, Hiroshima City Hiroshima Citizens Hospital, Hiroshima 730-8518, Japan; urokoba@gmail.com

**Keywords:** living donor kidney transplantation, urinary tract infection, perioperative, desensitization, rituximab, plasmapheresis, body mass index, dialysis duration, warm ischemic time, prophylactic antimicrobials

## Abstract

**Background/Objectives**: Limited research exists on risk factors for urinary tract infections (UTIs) in kidney transplant recipients, particularly in high-risk groups such as ABO-incompatible or donor-specific antibody (DSA)-positive cases. Early UTIs, especially within the first month post-transplant, impact on acute rejection and long-term graft outcomes, highlighting the need for risk factor identification and management. **Methods**: Among 157 living donor kidney transplant cases performed at our institution between 2009 and 2024, 128 patients were included after excluding cases with >72 h of perioperative prophylactic antibiotics or urological complications. UTI was defined as the presence of pyuria and a positive urine culture, accompanied by clinical symptoms requiring antibiotic treatment, occurring within one month post-transplantation. **Results**: The median onset of UTI was postoperative day 8 (interquartile range, IQR: 6.8–9.3). No subsequent acute rejection episodes were observed. The median serum creatinine at 1 month postoperatively was 1.3 mg/dL (IQR: 1.1–1.7), and this was not significantly different from those who did not develop UTI. In univariate analysis, low or high BMI (<20 or >25), longer dialysis duration (>2.5 years), desensitization therapy (plasmapheresis + rituximab), elevated preoperative neutrophil-to-lymphocyte ratio (NLR) (≥3), and longer warm ischemic time (WIT) (≥7.8 min) were significantly associated with an increased infection risk of UTI (*p* = 0.010, 0.036, 0.028, 0.015, and 0.038, respectively). Multivariate analyses revealed that abnormal BMI, longer dialysis duration, desensitization therapy, and longer WIT were independent risk factors for UTI (*p* = 0.012, 0.031, 0.008, and 0.033, respectively). The incidence of UTI increased with the number of risk factors: 0% (0/16) for zero, 10% (5/48) for one, 31% (16/51) for two, 45% (5/11) for three, and 100% (2/2) for four risk factors. **Conclusions**: Desensitization therapy, BMI, dialysis duration, and WIT were identified as independent risk factors for perioperative UTI. In patients with risk factors, additional preventive strategies should be considered, with extended antibiotic prophylaxis being one potential option.

## 1. Introduction

Urinary tract infection (UTI) is a risk factor for sepsis and acute rejection in kidney transplant recipients [1], and infections caused by gram-negative bacteria can cause acute antibody-related rejection [2,3]. UTI of the transplanted kidney, especially within the first month after surgery, has also been reported to affect renal function, graft loss, and overall survival [4].

Kidney transplantation is an open, clean-contaminated surgery that involves opening the urinary tract and using several types of immunosuppressive drugs (e.g., tacrolimus, mycophenolate mofetil, methyl-prednisone, mTOR inhibitor) before and after surgery; therefore, living donor recipients are considered at high risk for perioperative bacterial infection [5,6]. Although few reports exist regarding UTIs during the perioperative period, the rate is estimated to be 2–34% in the United States and Europe [6,7,8,9].

In Japan, living donor kidney transplants account for 90% of all transplants [10], of which ABO blood group incompatibility (ABO-I) accounts for as much as 30%.

Since many transplants are performed between parents and children as well as between married couples and other unrelated individuals, cases with anti-HLA antibodies (donor-specific antibodies; DSA) are not uncommon [11].

In Japan, immunologically high-risk cases are more prevalent compared to other countries [12]. As such, desensitization therapy combining plasmapheresis and rituximab has become the standard approach. Rituximab-based desensitization therapy for ABO-I transplant recipients has been reported to be associated with infectious complications [13]. Given this, we conducted this retrospective study to evaluate the potential impact of this treatment strategy on perioperative UTIs, as well as to explore other risk factors, in order to improve graft outcomes.

## 2. Materials and Methods

### 2.1. Patients (Recipients)

Individuals who underwent living donor kidney transplantation (LDKT) at Okayama University Hospital between 2009 and 2024 were included in this study. Since 2016, in accordance with the “Essential Japanese Guidelines for the Prevention of Perioperative Infections in the Urological Field: 2015 Edition” [14], single-dose prophylactic antimicrobial administration has been principally applied for kidney transplantation. (The guidelines recommend either a single-dose administration or continuation of antibiotics within 72 h postoperatively). The initial 24 patients whose administration period was longer than 72 h, which did not comply with the guidelines, and 4 patients who received antimicrobials unrelated to transplantation were excluded. In addition, one recipient with postoperative urinary complications was excluded because the recipient required additional surgery and was considered inappropriate for prophylactic antimicrobial therapy efficacy evaluation (Figure 1). We retrospectively collected data on recipient characteristics, manner of antimicrobial prophylaxis, application of desensitization therapy (defined in this study as the combined use of plasmapheresis and rituximab), and incidence of perioperative UTIs from patient medical records.

### 2.2. Perioperative UTI

The perioperative period was defined as within one month after LDKT. UTI was defined as a symptomatic infection accompanied by a positive culture requiring the administration of additional antimicrobial agents. Single-dose prophylactic antimicrobial agents administered for renal biopsy or removal of ureteral stents were not considered for the treatment of UTI. A positive culture was defined as bacteriuria (bacteria ≥ 1.0 × 10^4^ colony-forming units/mL) and the presence of pyuria (white blood-cell count ≥ 5/high-power fields) for UTI. A positive culture of a specimen from a patient requiring no additional antimicrobial administration was not counted as a perioperative UTI in this study.

### 2.3. Immunosuppressive Therapy Regimens

Figure 2 shows our institutional immunosuppressive therapy (IST) protocols. All recipients were given tacrolimus (calcineurin inhibitor; CNI), mycophenolate mofetil (MMF), and prednisolone preoperatively, with basiliximab administered before surgery on postoperative day (POD) 0 and on POD 4. For recipients with ABO identical blood types and ABO minor mismatch, IST was started 4 days before LDKT; it was started 1–2 weeks before LDKT for recipients with incompatible blood types. Double filtration plasmapheresis (DFPP) and plasma exchange (PEX) were performed within the week prior to LDKT in recipients with incompatible blood types (ABO major mismatch). The same protocol was applied regardless of isoagglutinin titers. A similar protocol was used for incompatible blood types for recipients with focal segmental glomerulosclerosis (FSGS). Immunosuppressants were administered 2–6 weeks before LDKT for recipients with DSAs; these recipients also underwent DFPP and PEX to reduce the level of DSAs. For patients at high risk, additional high-dose intravenous immunoglobulin (IVIG) was administered prior to LDKT. Low-dose rituximab (200 mg/body) was administered to all recipients, except those with identical blood types without DSAs and without FSGS, 7–14 days before LDKT.

### 2.4. Standard Procedures of LDKT

After hair was removed with surgical clippers and an indwelling 16 or 18 Fr urethral catheter was placed, the skin was prepared with povidone-iodine, and a Gibson incision was made via a retroperitoneal approach. The lymphatic vessels around the pertinent blood vessels were ligated or cut with an electrosurgical knife. End-to-side anastomoses of the renal vessels to the external iliac vein and artery and ureterovesical anastomoses were made with the Lich–Gregoir technique. All recipients were given 4.7–5 Fr 10 cm double-J ureteral stents and retroperitoneal drains (Pleats Drainage Tube 8 mm in outside diameter). Wounds were closed with interrupted sutures for the fascia and subcutaneous fat and with running subcuticular absorbable sutures for the skin. Subcutaneous drains (Blake 10 or 15 Fr) were placed in recipients with at least 2 cm of subcutaneous fat; these were removed when drainage decreased to less than 10 mL per day. Urethral catheters were usually removed after 1 week, and ureteral stents were removed within 2–3 weeks after LDKT. In our institution, warm ischemic time (WIT) is defined not as the period until the graft is placed on ice for cooling, but rather as the interval from the opening of the graft renal vessels on the back table to the initiation of perfusion with preservation solution.

### 2.5. Bacteriological Examination and Prophylactic Antimicrobial Agents

Urine cultures were obtained from all recipients before LDKT, except for recipients with anuria. Those with negative urine cultures and anuria were administered prophylactic ampicillin/sulbactam 750 mg or cefazolin (CEZ) 500 mg once daily without additional administration intraoperatively. For recipients with a positive urine culture, prophylactic antimicrobials were selected according to the results of culture and susceptibility testing. These medications were administered over the same timeframe as for recipients with negative urine cultures. Weekly urine cultures were performed after LDKT (not performed without pyuria or bacteriuria), and the tips of the retroperitoneal and subcutaneous drains and intravascular catheters were cultured at removal. At endoscopic ureteral-stent removal, a single dose of a prophylactic antimicrobial agent was administered. Oral fluoroquinolones (e.g., levofloxacin) were usually used, and if the postoperative urine culture was positive, the antibiotic was selected accordingly. A single prophylactic dose of CEZ was administered at the time of renal biopsy.

### 2.6. Statistical Methods and Analyses

All analyses were conducted using EZR (Saitama Medical Center, Jichi Medical University, Saitama, Japan) [15]. For comparisons between two groups, a Chi-square test or Fisher’s exact test was used as appropriate. Continuous variables with normal and non-normal distribution were analyzed with Student’s *t*-test and the Mann–Whitney U test, respectively. The Youden index was referred to as a guide when converting continuous variables into binary variables for analysis. In univariate and multivariate analyses, logistic regression analyses were used to determine independent risk factors for perioperative UTI after LDKT. Variables that showed statistical significance in the univariate analysis were included in the multivariate model. A *p*-value of <0.05 was considered statistically significant.

### 2.7. Ethics

This retrospective study was approved by the Ethics Committee of Okayama University Hospital (registration no. 1507-003).

## 3. Results

### 3.1. Recipient Characteristics

The recipient characteristics and preoperative conditions are summarized in Table 1. Diabetes Mellitus (DM), also known to be a risk factor for surgical site infection (SSI), was present in 27 recipients (21%). ABO-I was found in 49 recipients (38%), DSA positivity was found in 25 (20%), including Flow Cytometry Crossmatch positivity with high-dose IVIG administered in 2 (2%). Desensitization therapy (plasmapheresis + rituximab) was applied to 74 (58%), and a positive preoperative urine culture was observed in 20 recipients (16%).

### 3.2. Antimicrobial Prophylaxis and Results of the Procedure

Table 2 shows the prophylactic antimicrobial agents administered and the details of the procedure. A positive urine culture before LDKT was observed in 20 recipients, whereas there was no recipient with symptomatic UTI. Thus, 111 recipients were given cefazolin (87%), and 9 recipients were given ampicillin/sulbactam (7%).

Based on preoperative urine culture results, cefotiam (CTM) or cefmetazole (CMZ) was administered to one patient each, and flomoxef (FMOX) or levofloxacin (LVFX) was administered to three patients each. CTM was used in a recipient in whom quinolone-resistant Escherichia coli was identified, and CMZ was selected for a case with extended-spectrum beta-lactamase (ESBL)-producing *E. coli*. FMOX was administered to recipients with ESBL-producing *E. coli*, while LVFX was chosen due to either a cephalosporin allergy or detection of Enterococcus faecalis or Serratia marcescens.

Although urethral catheters and retroperitoneal drains were placed in all recipients, a subcutaneous drain was indicated in 37 recipients (29%). The median durations of urethral catheter, retroperitoneal drain, and subcutaneous drain placement were 7, 5, and 6 days, respectively.

### 3.3. Perioperative UTIs and Outcomes

A total of 28 patients (22%) received additional antibiotic treatment for UTI (Table 2). Fever was observed in 25 patients (89%), while others exhibited bladder irritation and systemic symptoms, including nausea, fatigue, etc. Details of each factor in the UTI-positive and UTI-negative groups are summarized in Table 3. The bacterial species isolated from urine cultures are summarized in Table 4.

A total of 31 bacterial strains were identified in the UTI group, of which 29 (94%) were gram-negative organisms, detected in 26 recipients (93%). Blood cultures were positive in 3 patients (2% of the total cohort; 11% of those with UTI), with *E. coli* (non-ESBL-producing) isolated in all cases. The administered antibiotics included CTM, LVFX, and piperacillin-tazobactam (PIPC/TAZ).

All patients who received additional antibiotics for UTI recovered without the need for further invasive interventions, and all indwelling catheters, including ureteral stents, were removed. No cases of perioperative acute rejection following UTI were observed.

One-year patient and graft survival rates were 100%, regardless of the presence or absence of UTI.

### 3.4. Risk Factors for Perioperative UTIs

Table 5 presents the results of a univariate analysis evaluating perioperative risk factors for UTI. A trend toward increased incidence of UTI was observed in patients with either low or high body mass index (BMI); thus, BMI was recategorized as a categorical variable (low or high BMI: <20 or >25). Longer dialysis duration, elevated preoperative neutrophil-to-lymphocyte ratio (NLR), and longer WIT also showed similar trends, and the cutoff values were set: dialysis duration > 2.5 years, NLR ≥ 3, and WIT ≥ 7.8 min. None of the potential risk factors, including age, sex, diabetes mellitus (DM), and preoperative positive urine culture, showed a statistically significant difference [16]. However, patients who received prophylactic CEZ tended to have a higher incidence of UTI (*p* = 0.12). Statistically significant associations were observed for the following factors: preoperative variables including low or high BMI, longer dialysis duration, desensitization therapy, and elevated NLR, as well as longer WIT as another relevant factor (*p* = 0.010, 0.036, 0.015, 0.028, and 0.038, respectively).

The results of the multivariate analysis are presented in Table 6. In the model including only preoperative factors (Model I), BMI, dialysis, and desensitization therapy were identified as significant variables, with odds ratios (ORs) of 3.0 (95% confidence interval [CI], 1.18–7.63; *p* = 0.021), 2.88 (95% CI, 1.04–8.0; *p* = 0.042), and 3.54 (95% CI, 1.24–10.1; *p* = 0.018), respectively. The area under the curve (AUC) for Model I was 0.759 (95% CI, 0.665–0.854). Furthermore, in the analysis of Model II, in which WIT was included as the intraoperative factor, BMI, dialysis duration, desensitization therapy, and WIT were identified as independent risk factors, with ORs of 3.35 (95% CI, 1.3–8.6; *p* = 0.012), 3.15 (95% CI, 1.11–8.96; *p* = 0.031), 4.31 (95% CI, 1.46–12.8; *p* = 0.008), and 3.53 (95% CI, 1.1–11.3; *p* = 0.033), respectively. The AUC for Model II was 0.764 (95% CI, 0.672–0.856).

In this study (Figure 3), the incidence of UTI increased with the number of risk factors present: 0% (0/16) in patients with no risk factors, 10% (5/48) with one, 31% (16/51) with two, 45% (5/11) with three, and 100% (2/2) with four risk factors. The trend was statistically significant (Cochran–Armitage trend test, *p* < 0.00001).

## 4. Discussion

A In this study, we explored the risk factors of perioperative UTI in LDKT, with a focus on the impact of desensitization therapy. We found that desensitization therapy itself, BMI, dialysis duration, and WIT were independent risk factors for the occurrence of perioperative UTI. Perioperative infectious complications in kidney transplant recipients have been associated with several risk factors, including age, BMI, ASA classification, DM, and serum albumin levels [17,18]. Several factors have been reported to be associated with the occurrence of SSI, including impaired wound healing and complications. These include anemia, expanded criteria donors (ECD), delayed graft function (DGF), prolonged ischemia time, the presence of urological complications, and the use of steroids or mTOR inhibitors [19]. Reports not limited to the perioperative period (1 month) have identified age, female sex, prolonged urinary catheterization (over 7 days), and use of ureteral stents as risk factors for UTI [1]. In LDKT, characterized by a high prevalence of desensitization therapy, we examined the risk factors identified in this study and explored potential strategies to reduce the incidence of perioperative UTIs as follows.

While some studies report rituximab as a risk factor for postoperative bacterial infections [20,21], others suggest no significant difference between treated and untreated groups [22]. Plasmapheresis has been reported as a potential risk factor for infections in kidney transplant recipients [23,24]. Chung et al. [25] reported a significant increase in post-transplant infection risk with combined desensitization therapy involving rituximab and plasmapheresis. Generally, immunologically high-risk recipients undergoing desensitization therapy (e.g., ABO-incompatible or DSA-positive cases) receive a combination of CNI, MMF, and steroids prior to transplantation, which may synergistically increase susceptibility to infections. In this study, among 16 low-immunological-risk cases who received rituximab without plasmapheresis, only one developed UTI, and no other infections were observed.

Obesity is a known perioperative risk factor for SSI under immunosuppressive conditions, and SSIs are associated with significantly lower graft survival rates [26]. However, in our study, no SSIs requiring treatment were observed. This may be attributed to the placement of subcutaneous drains in recipients with subcutaneous fat thickness exceeding 2 cm (median BMI 26.3 for drain placement cases vs. 20.4 for non-placement). In most cases (87%), CEZ, the first-line agent for SSI prevention, was used as the perioperative prophylactic antibiotic. Among the general population, increased BMI has been associated with higher risks of UTI and pyelonephritis [27], while in chronic kidney disease (CKD), low BMI (≤20) has been reported as a risk factor for infection-related mortality [28,29]. Although the definition of low BMI as <20 and high BMI as ≥25 in this study may be debatable, it is notable that Asians, including Japanese individuals, generally have higher body fat percentages at the same BMI compared with Western populations [30]. The Japan Society for the Study of Obesity defines obesity as BMI ≥ 25 [31], in contrast to the WHO definition of BMI ≥ 30 [32], supporting the appropriateness of these cutoffs for this population. Furthermore, recent attention has been given to sarcopenia and frailty, which are associated with postoperative infection risks across various fields. These conditions are linked to impaired immunity due to malnutrition [33,34]. Although the diagnosis of sarcopenia can be complex, our study suggests that simple BMI measurement may provide a practical perioperative risk assessment tool in clinical settings.

Prolonged dialysis period is known to reduce bladder capacity and compliance [35]. Hotta et al. reported that among patients on dialysis for over 10 years, 56 of 99 (56%) had atrophic bladders with capacity ≤ 50 mL, posing a risk for postoperative urinary complications and vesicoureteral reflux (VUR) [36]. In our study, median bladder capacity among 28 patients with UTIs was 200 mL (IQR: 143–300), and no urinary complications or VUR were noted. Measurements were based on intraoperative saline infusion volumes during ureteroneocystostomy or maximum voided volume during hospitalization. On the other hand, reports suggest that dialysis may impair bladder mucosal barrier function, increasing susceptibility to infections [37]. The finding that dialysis duration over 30 months (2.5 years) was a risk factor for UTI in our study implies that barrier dysfunction may precede notable volume reduction. In this study, 29 patients had positive urine cultures postoperatively but did not require antibiotic treatment. Among them, desensitization therapy had been administered in 18 cases (62%). Approximately half of the isolates were gram-negative bacteria (Appendix A). Even in the context of frequent use of desensitization therapy, a shorter dialysis duration may contribute to preserved mucosal barrier function, potentially reducing the risk of developing UTI. The median duration of dialysis in this group was 6 months (interquartile range [IQR]: 0–14 months). However, this decline in barrier function may also be influenced by age, general health status, DM, and duration of anuria.

According to a previous report [36], only 2% of patients on dialysis for less than 5 years and 8% on dialysis for less than 10 years had atrophic bladders (≤50 mL), suggesting that routine bladder capacity evaluation via cystography may be omitted in short- to mid-term patients on dialysis. Since a history of pre-transplant UTI is a known risk factor for post-transplant UTI [38], cystography, being potentially invasive, should be performed selectively.

The NLR has recently gained attention as a marker of immune response and inflammation in various fields, including cancer immunotherapy prognosis, cardiovascular event prediction, and COVID-19 severity assessment. It is inexpensive and easily calculated from standard blood counts. A high preoperative NLR has been reported as a risk factor for perioperative infectious complications in patients with gynecological and gastric cancers [39,40], as well as other gastrointestinal and renal cancers. However, few studies have assessed perioperative UTI risk using NLR in CKD or kidney transplant recipients. In our multivariate analysis, NLR was not identified as a risk factor (*p* = 0.129). A previous study on prostate cancer [41] reported that a significant postoperative increase in NLR from preoperative levels correlated with infection outcomes. Considering that NLR may fluctuate with preoperative rituximab administration, MMF, and steroid use, we plan to include immediate pre-transplant NLR measurements in future studies.

Longer WIT has been reported as a risk factor for DGF [42], and DGF itself has also been associated with an increased risk of UTI [43]. However, to the best of our knowledge, no previous studies have directly identified longer WIT as a risk factor for perioperative UTI. Experimental studies using animal models have suggested that extended ischemia may compromise renal anti-infective defense mechanisms due to ischemia–reperfusion injury [44]. The present findings may indicate that a similar mechanism contributes to increased susceptibility to infection in human renal allografts. Further histological investigations are warranted to clarify this mechanism.

In this study, the incidence of UTI tended to increase linearly with the number of risk factors. Notably, patients with three or more risk factors developed UTIs at a high rate of 54% (7/13). However, UTI did not progress to a severe infection requiring ICU management, such as DIC or septic shock, and no secondary acute rejection episodes were observed. Kidney function at one month postoperatively showed no difference between the UTI and non-UTI groups. These results may reflect prompt initiation of treatment based on routine symptom assessments, vital signs, blood and urine tests, and culture examinations. Nonetheless, strategies for UTI prevention in high-risk patients remain a challenge.

Among the 28 patients who developed UTI, *E. coli* was detected in 19 cases (68%), with quinolone-resistant or ESBL-producing strains identified in 9 cases. Overall, over 90% of isolates were Gram-negative bacteria. While the single-dose perioperative prophylactic use of CEZ, recommended by the Japanese, EAU [45], and AUA [46] guidelines, was not a significant risk factor for UTI (*p* = 0.161), the limited number of patients who received antibiotics other than CEZ (n = 17) may have resulted in a lack of statistical power. From a coverage perspective, second-generation cephalosporins or beta-lactamase inhibitor combinations may be effective in preventing perioperative UTI. However, CEZ may still be superior in preventing SSI, warranting further comparative investigation. From the perspective of preventing SSI, CEZ may be considered more favorable. However, our department has previously reported retrospective findings in robot-assisted radical prostatectomy, indicating that the risk of postoperative infections—including UTI, SSI, and remote infection (RI)—did not differ significantly based on the type of antibiotic (CEZ vs. ABPC/SBT) or the duration of administration [1]. Therefore, prospective studies for prophylactic antibiotics are warranted in the perioperative management of LDKT.

Empirical use of broad-spectrum antibiotics (e.g., carbapenems, TAZ/PIPC) is common when UTI occurs post-transplant. Targeted prophylactic antibiotics covering expected pathogens may help reduce antimicrobial resistance (AMR) and cost. Rodrigo et al. reported that fosfomycin prophylaxis during urinary catheter or stent removal significantly reduced UTI [47]. The AUA also recommends considering single-dose prophylaxis at catheter removal in high-risk patients, such as transplant recipients [46].

In our study, the median UTI onset was POD 8 (IQR 6.8–9.3). Among UTI cases, 16 patients (57%) developed infection after catheter removal, and 8 patients (29%) developed UTI before removal (postponing removal). Four patients (14%) developed UTI during catheter retention beyond 8 days. Although it was not known whether prophylaxis was administered at catheter removal, previous literature indicates that the median UTI onset was 19 days (IQR 9–40) and 62 days (range 11–205) [48,49]. Compared with these reports, UTI onset in our high-risk cohort was notably earlier. Preventive measures, such as administering antibiotics on days 5–7 or single-dose prophylaxis at catheter removal, may reduce UTI incidence. If no complications like urine leakage exist, earlier catheter removal combined with single-dose prophylaxis might be ideal. Based on our prior studies [50], effective options against resistant *E. coli* and other Gram-negative bacteria may include FMOX, CMZ, or oral sitafloxacin.

A noteworthy outcome of this study is that both the 1-year patient and graft survival rates were 100%. Only one patient required additional treatment for an anastomotic complication that was diagnosed more than one year postoperatively and treated with Deflux injection for VUR. However, the incidence of perioperative UTI was 22%, which is not considered low.

To reduce the risk of infection-related deterioration in graft function, we are currently considering revising our treatment algorithm for kidney transplant recipients, incorporating additional prophylactic antibiotic strategies for perioperative UTI high-risk patients, based on the findings of this study (Appendix A). Furthermore, recent studies have reported the development of robust diagnostic and predictive models for UTIs using risk scores. In the future, we aim not only to accumulate additional cases at our institution but also to establish a UTI risk score after kidney transplantation through larger, multicenter collaborative studies [51,52,53].

Limitations of this study include the following: (1) The retrospective, single-center, non-randomized nature with a limited sample size—prospective studies with larger cohorts are needed; (2) The potential confounders, including demographic characteristics (e.g., age and sex) and comorbidities such as obesity and diabetes mellitus, were not systematically adjusted for in the analysis. These factors are known to affect infection risk, and their omission may have partially influenced our results. Future studies with larger cohorts and comprehensive adjustment for these confounders are warranted: (3) Desensitization protocols vary based on immunologic risk, recipient condition, and specifics of rituximab, PE type/frequency, potentially impacting UTI risk differently; (4) The use of prophylactic trimethoprim-sulfamethoxazole, TMP-SMX (for Pneumocystis jirovecii pneumonia prevention starting at week 2), single-dose quinolones at stent removal, and CEZ at biopsy may have suppressed UTI incidence. However, many isolated pathogens showed resistance to TMP-SMX, as also noted in previous reports [42].

## 5. Conclusions

We found that desensitization therapy, low or high BMI, longer dialysis duration, and longer WIT were significantly associated with an increased risk of perioperative UTI in LDKT. Individualized preventive strategies based on the number of risk factors, including the selective use of additional prophylactic antibiotics, should be considered for high-risk patients.

## Figures and Tables

**Figure 1 jcm-14-06102-f001:**
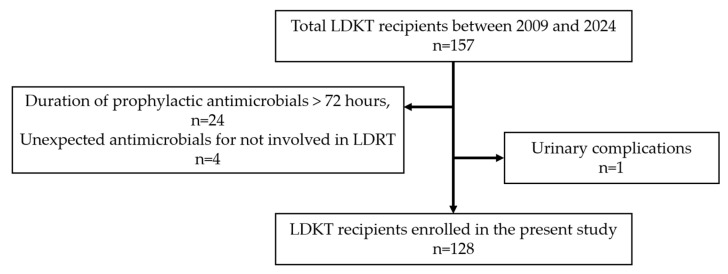
Recipient flow in the present study.

**Figure 2 jcm-14-06102-f002:**
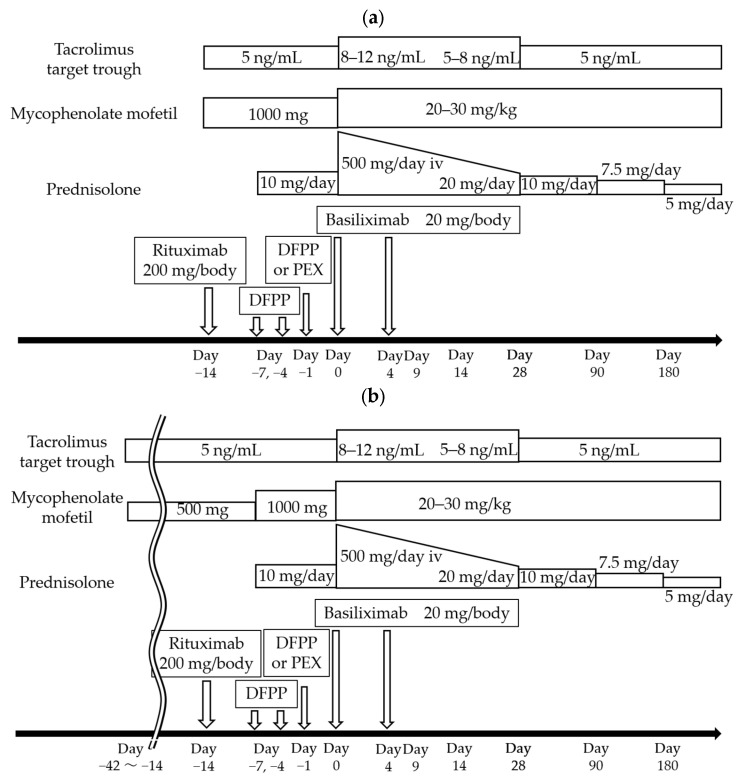
Immunosuppressive therapy (IST) regimens: (**a**) For ABO major mismatch (ABO incompatible); (**b**) For donor-specific antibody (DSA) positive living-donor kidney transplant recipients.

**Figure 3 jcm-14-06102-f003:**
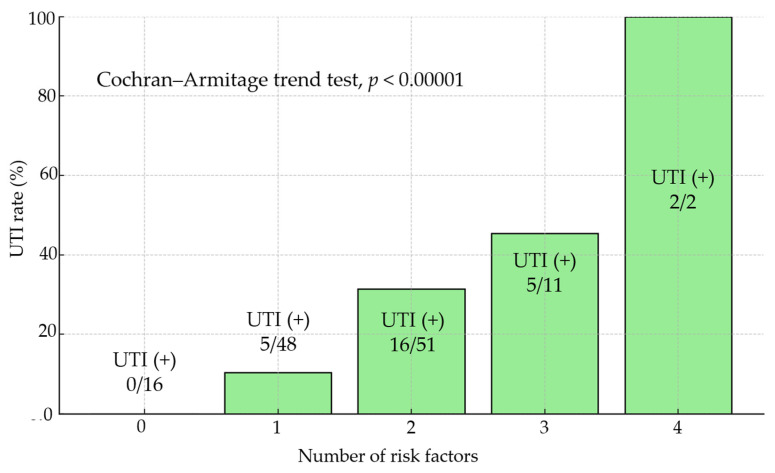
UTI rates by number of risk factors.

**Table 1 jcm-14-06102-t001:** Patient characteristics.

N	128	
Median age (IQR) ^a^	43	(32.8–57)
Median BMI ^b^, kg/m^2^ (IQR)	21.7	(19.2–24.4)
Median duration of dialysis before LDKT ^c^ (n = 91), year (IQR)	1.3	(0.5–3)
Pre-LDKT s-albumin, ^d^ g/dL (IQR)	3.9	(3.5–4.1)
Pre-LDKT Neutrophil-to-lymphocyte ratio (NLR), (IQR)	2.7	(2–3.8)
Sex, n (%)		
male	86	(67)
female	42	(33)
ASA ^e^ physical status classification, n (%)		
2	5	(4)
3	119	(93)
4	4	(3)
Diabetes Mellitus, n (%)	27	(21)
Diabetic nephropathy, n (%)	24	(19)
ABO-incompatible, n (%)	49	(38)
Donor-specific antibody, n (%)	25	(20)
Administration of rituximab, n (%)	90	(70)
Plasmapheresis + rituximab before LDKT, n (%)	74	(58)
Pre-emptive kidney transplantation, n (%)	37	(29)
Positive urine culture before surgery, n (%)	20	(16)
Donor		
Median age (IQR)	60	(53–67)
Expanded criteria donor, n (%)	69	(54)

^a^ IQR, interquartile range; ^b^ BMI, body mass index; ^c^ LDKT, living donor kidney transplantation; ^d^ s-. albumin, serum-albumin; ^e^ ASA, American Society of Anesthesiologists.

**Table 2 jcm-14-06102-t002:** Details and procedures of antimicrobial prophylaxis and Results.

N		128	
Type of prophylactic antimicrobials, cases (%)			
	Ampicillin/sulbactam	9	(7)
	Cefazolin	111	(87)
	Cefotiam	1	(1)
	Cefmetazole	1	(1)
	Flomoxef	3	(2)
	Levofloxacin	3	(2)
Duration of prophylactic antimicrobials, cases (%)			
	Single-dose	86	(67)
	Within 72 h (up to POD ^a^ 2)	42	(33)
Median operative time, min (IQR)		490	(427–559)
Median EBL ^b^, mL (IQR)		178	(100–318)
Median WIT ^c^, min (IQR)		5	(3.3–6.7)
With retroperitoneal drain, cases (%)		128	(100)
	Median duration of placement, days (IQR)	5	(4–7)
Urethral catheter, cases (%)		128	(100)
	Median duration of placement, days (IQR)	7	(7–8)
Subcutaneous drain placement, cases (%)		37	(29)
	Median duration of placement, days (IQR)	6	(4–9)
Surgical site infection, n (%)		0	(0)
Urinary tract infection, n (%)		28	(22)
Lymphocele, n (%)		5	(4)
Viral infection, n (%)		2	(2)
Fungal infection, n (%)		1	(1)
Delayed graft function, n (%)		1	(1)
Acute rejection within 1-month, n (%)		2	(2)
Graft loss within 1-month, n (%)		0	(0)
1-month postoperative serum-Creatinine, mg/dL (IQR)		1.3	(1.1–1.7)

^a^ POD, postoperative day; ^b^ EBL, estimated blood loss; ^c^ WIT, warm ischemic time.

**Table 3 jcm-14-06102-t003:** Factor details in UTI (+) and UTI (−) groups.

	UTI (−)		UTI (+)	
N	100		28	
Median age (IQR)	47	(31–57)	44.5	(36.8–57.3)
Median age of donor (IQR)	59.5	(53–66.3)	62.5	(54.3–68)
Median BMI, kg/m^2^ (IRQ)	21.7	(19.5–24.3)	21.1	(18.2–25.6)
Median duration of dialysis before LDKT (n = 91), years (IQR)	0.5	(0–1.7) n = 68	1.3	(0.2–3) n = 23
Median pre-LDKT s-albumin, mg/dL (IQR)	3.9	(3.5–4.2)	3.8	(3.5–4.1)
Median pre-LDKT NLR, (IQR)	2.5	(2–3.6)	3.3	(2.1–4)
Sex, n (%)				
male	68	(68)	18	(64)
female	32	(32)	10	(36)
ASA risk, n (%)				
2	5	(5)	0	(0)
3	93	(93)	26	(93)
4	2	(2)	2	(7)
Diabetic Mellitus, n (%)	19	(19)	8	(29)
ABO incompatible LDKT, n (%)	35	(35)	14	(50)
Donor-specific antibody, n (%)	15	(15)	9	(32)
Administration of rituximab, n (%)	67	(67)	23	(82)
Plasmapheresis + rituximab before LDKT, n (%)	52	(52)	22	(79)
Pre-emptive kidney transplantation, n (%)	32	(32)	5	(18)
Positive urine culture before surgery, n (%)	17	(17)	3	(11)
Donor				
Median age (IQR)	59.5	(53–66.3)	62.5	(54.3–68)
Expanded criteria donor, n (%)	51	(51)	18	(64)
Type of antimicrobials, n (%)				
cefazolin	84	(84)	27	(96)
ampicillin/sulbactam, FMOX, LVFX, CTM, CMZ	16	(16)	1	(4)
Duration of prophylactic antimicrobials, n (%)				
Single-dose	65	(65)	21	(75)
Within 72 h (up to POD2)	35	(35)	7	(25)
Median operative time, min (IQR)	493	(442–577)	467	(404–555)
Median EBL, mL (IQR)	190	(100–333)	145	(100–224)
Median WIT, min (IQR)	5	(3.3–6.5)	5.1	(3–7.9)
Median duration of pelvic drain placement, days (IQR)	5	(4–7)	5	(4–8)
Median duration of urethral catheter placement, days (IQR)	7	(7–7)	7	(7–13)
Median duration of subcutaneous drain placement (n = 37), days (IQR)	5.5	(4–8.3) n = 29	7.5	(4.5–9.3) n = 8

**Table 4 jcm-14-06102-t004:** Profiles of pathogens isolated postoperatively.

Urine Samples	Number of Isolates
*Escherichia coli*	10
*Escherichia coli* (QREC ^a^)	4
*Escherichia coli* (ESBL ^b^)	5
*Klebsiella pneumoniae*	3
*Enterobactor cloacae*	3
*Proteus mirabilis*	2
*Pseudomonas aeruginosa*	1
*Citrobactor speies*	1
CNS ^c^	2

^a^ QREC, Quinolone-resistant Escherichia coli; ^b^ ESBL, Extended-spectrum β-lactamase; ^c^ CNS: Coaglase-negative staphyococci.

**Table 5 jcm-14-06102-t005:** Univariate analysis of risk factors for UTI.

	Univariate Analysis		
Variable	OR ^a^	95% CI ^b^	*p*
Age (<45)	0.85	0.37–1.97	0.708
Sex (Female)	1.18	0.49–2.85	0.712
BMI (<20, or 25<, kg/m^2^)	3.14	1.32–7.46	0.01
Duration of dialysis before LDKT (2.5<, years)	2.71	1.07–6.89	0.036
Serum-albumin (<3.9, g/dL)	3.9	0.35–1.85	0.603
ASA (3<)	3.77	0.51–28.1	0.195
Diabetes Mellitus	1.71	0.65–4.45	0.276
Pre-LDKT NLR (3≤)	2.63	1.11–6.22	0.028
Administration of rituximab	2.27	0.79–6.49	0.128
Desensitization (plasmapheresis + rituximab)	3.38	1.26–9.06	0.015
Positive urine culture before LDKT	0.59	0.16–2.16	0.422
Administration of cefazolin	5.14	0.65–40.6	0.12
Duration of prophylactic antimicrobials (single-dose)	0.62	0.24–1.6	0.322
Donor Age (60<)	2.11	0.87–5.11	0.098
Expanded criteria donor	1.73	0.73–4.11	0.215
WIT (7.8≤, min)	2.93	1.06–8.12	0.038
Duration of retroperitoneal drain placement (10<, days)	1.45	0.47–4.5	0.515
Duration of urethral catheter placement (7<, days)	0.86	0.31–2.38	0.777
Duration of double-J ureter stent placement (18<, days)	0.59	0.23–1.53	0.279

^a^ OR, odds ratio; ^b^ 95% CI, 95% confidence interval.

**Table 6 jcm-14-06102-t006:** Multivariate logistic regression analyses for perioperative UTI (Model I, II).

		Multivariate Analysis		
	Variable	OR	95% CI	*p*
**Model I**	BMI (<20, or 25<, kg/m^2^ )	3	1.18–7.63	0.021
	Duration of dialysis before LDKT (2.5<, years)	2.88	1.04–8	0.042
	Desensitization (plasmapheresis + rituximab)	3.54	1.24–10.1	0.018
	Pre-LDRT NLR (3≤)	2.06	0.82–5.21	0.126
	AUC = 0.759		0.665–0.854	
**Model II**	BMI (<20, or 25<, kg/m^2^ )	3.35	1.3–8.6	0.012
	Duration of dialysis before LDKT (2.5<, years)	3.15	1.1–9	0.031
	Desensitization (plasmapheresis + rituximab)	4.31	1.46–12.8	0.008
	WIT (7.8≤, min)	3.53	1.1–11.3	0.033
	AUC = 0.764		0.672–0.856	

## Data Availability

Data supporting the findings of this study are available from the corresponding author on reasonable request.

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
