# Peer review of "Risk Factors for Perioperative Urinary Tract Infection After Living Donor Kidney Transplantation Characterized by High Prevalence of Desensitization Therapy: A Single-Center Analysis"

_jcm, 2025, doi:10.3390/jcm14176102_

Round 1

Reviewer 1 Report

Comments and Suggestions for Authors

This is a very well conducted study from Dr. Nishimura et al. The topic is very important as the population living with kidney transplant is rising. My suggestion are included as below. 

2. Methods: you might want to clarify that your multivariable models were created using the significant variables in the univariable analysis. By doing this, the readers might understand your paper easier. 

3. Results:

3.1 I suggest including a table showing the univariable analysis for differences between UTI vs non-UTI. Table 4 is good but I prefer to see the actual number and percentages.

3.2 Moreover, an AUC should be provided with its 95% CI. 

4. Discussion

4.1 Could you include a brief discussion about the potential of future risk prediction model for UTI after LDKT? Many risk scores for UTI in different scenario (outpatient, inpatient) had been developed and partially validated ( https://doi.org/10.1371/journal.pone.0323664 ; https://doi.org/10.1371/journal.pone.0290215 ; https://doi.org/10.1371/journal.pone.0248636 ) You might consider to discuss about it and those papers can be used for your references. I understand that the sample size will be very challenging but kidney transplant population is rising. 

4.2 You might want to mention in your limitation that potential confounders (comorbidities) exist and had not been addressed. 

Author Response

Comments 1: you might want to clarify that your multivariable models were created using the significant variables in the univariable analysis. By doing this, the readers might understand your paper easier.

Response 1: Thank you for pointing this out. I agree with comment. In accordance with your suggestion, I have inserted “Variables that showed statistical significance in the univariate analysis were included in the multivariate model.” on page 5, line 167.

Comments 2: I suggest including a table showing the univariable analysis for differences between UTI vs non-UTI. Table 4 is good but I prefer to see the actual number and percentages.

Rssponse 2: Thank you for pointing this out. I agree with comment. In accordance with your suggestion, I have inserted new Table on page 7, line 222.

Comments 3: Moreover, an AUC should be provided with its 95% CI.

Rssponse 3: Thank you for pointing this out. I agree with comment. In accordance with your suggestion, I have inserted 95% CI on page 10 (table).

Comments 4: Could you include a brief discussion about the potential of future risk prediction model for UTI after LDKT? Many risk scores for UTI in different scenario (outpatient, inpatient) had been developed and partially validated ( https://doi.org/10.1371/journal.pone.0323664 ; https://doi.org/10.1371/journal.pone.0290215 ; https://doi.org/10.1371/journal.pone.0248636 ) You might consider to discuss about it and those papers can be used for your references. I understand that the sample size will be very challenging but kidney transplant population is rising. 

Rssponse 4: Thank you for pointing this out. I agree with comment. In accordance with your suggestion, I have inserted "Furthermore, recent studies have reported the development of robust diagnostic and predictive models for UTIs using risk scores. In the future, we aim not only to accumulate additional cases at our institution but also to establish a UTI risk score after kidney transplantation through larger, multicenter collaborative studies [51-53]" on page 13, line 401. I have added the three references to the References section on page 17, line 565.

Comments 5: You might want to mention in your limitation that potential confounders (comorbidities) exist and had not been addressed. 

Rssponse 5: Thank you for pointing this out. I agree with comment. In accordance with your suggestion, I have inserted "(2) The potential confounders, including demographic characteristics (e.g., age and sex) and comorbidities such as obesity and diabetes mellitus, were not systematically adjusted for in the analysis. These factors are known to affect infection risk, and their omission may have partially influenced our results. Future studies with larger cohorts and comprehensive adjustment for these confounders are warranted" on page 13, line 408.

Reviewer 2 Report

Comments and Suggestions for Authors

The authors present a very well-prepared cohort of living kidney transplants. The data is presented in a comprehensible manner and provides considerable added value for the reader. The depth of detail in the admittedly small cohort is particularly convincing, especially given the immunologically complex transplant profile. The results are not particularly surprising, but the good preparation and presentation are positive.

Overall, a very pleassant manuscript with minor comments.

The title is a weak point of the work. The addition:  ... "characterized by high prevalence of desensitization therapy" seems unnecessary to me; I would consider it more important to emphasize that this is a single-center analysis. 

Minor points:

Abstract:

The abstract is too long; please shorten it appropriately. In particular, the detailed presentation of the results is excessive.

The sentence: „Desensitization therapy (plas- 25 mapheresis + rituximab) was performed in 74 cases (58%) for ABO-incompatible or DSA- 26 positive transplantation.“ Belongs tot he results part not to methods.

Introduction:

„Since many transplants are performed between parents and children as well as between married couples and other unrelated individuals, cases with anti-HLA antibodies (donor specific antibodies; DSA) are not uncommon.“ This sentences needs to be paralleled by numbers and a reference.

Methods:

Is the presented protocol valid for all patients, regardless of the existing isogglutinin titers in ABO-incompatible patients? Or are there different protocols?

Results:

The warm ischemia time seems very short. How was this defined? Please include this in the methods section.

Author Response

Comments 1: The title is a weak point of the work. The addition:  ... "characterized by high prevalence of desensitization therapy" seems unnecessary to me; I would consider it more important to emphasize that this is a single-center analysis. 

Response 1: Thank you for pointing this out. As our study demonstrated that desensitization therapy itself was identified as a risk factor for UTI, we have revised the title accordingly. In addition, following your suggestion, we have included the wording “A Single-Center Analysis” in the title to clarify the nature of our study design.

Comments 2: The abstract is too long; please shorten it appropriately. In particular, the detailed presentation of the results is excessive.

The sentence: „Desensitization therapy (plas- 25 mapheresis + rituximab) was performed in 74 cases (58%) for ABO-incompatible or DSA- 26 positive transplantation.“ Belongs tot he results part not to methods.

Rssponse 2: Thank you for pointing this out. I agree with comment. In accordance with your suggestion, I have shortened the abstract by deleting the indicated “Desensitization therapy (plas- 25 mapheresis + rituximab) was performed in 74 cases (58%) for ABO-incompatible or DSA- 26 positive transplantation.” as well as some of the detailed analytical results.

Comments 3: „Since many transplants are performed between parents and children as well as between married couples and other unrelated individuals, cases with anti-HLA antibodies (donor specific antibodies; DSA) are not uncommon.“ This sentences needs to be paralleled by numbers and a reference.

Rssponse 3: Thank you for pointing this out. I agree with comment. In accordance with your suggestion, I have inserted reference[11] on page 2, line 64.

"Japanese Society for Clinical Renal Transplantation, The Japan Society for Transplantation. “Annual Progress  Report  from  the  Japanese  Renal  Transplant  Registry:  Number  of  Renal  Transplantations  in  2023  and  Follow-up  Survey”. Available from: https://doi.org/10.11386/jst.59.3_217."

Comments 4: Is the presented protocol valid for all patients, regardless of the existing isogglutinin titers in ABO-incompatible patients? Or are there different protocols?

Rssponse 4: Thank you for pointing this out. At present, in our institution, the protocol for desensitization therapy does not differ according to titers. I have inserted "The same protocol was applied regardless of isoagglutinin titers." on page 3, line 110.

Comments 5: The warm ischemia time seems very short. How was this defined? Please include this in the methods section.

Rssponse 5: Thank you for pointing this out. I have inserted "In our institution, warm ischemic time (WIT) is defined not as the period until the graft is placed on ice for cooling, but rather as the interval from the opening of the graft renal vessels on the back table to the initiation of perfusion with preservation solution." on page 4, line 139.